# The Prevalence of Tick-Borne Encephalitis Virus in Wild Rodents Captured in Tick-Borne Encephalitis Foci in Highly Endemic Lithuania

**DOI:** 10.3390/v16030444

**Published:** 2024-03-13

**Authors:** Evelina Simkute, Arnoldas Pautienius, Juozas Grigas, Marina Sidorenko, Jana Radzijevskaja, Algimantas Paulauskas, Arunas Stankevicius

**Affiliations:** 1Laboratory of Immunology, Department of Anatomy and Physiology, Lithuanian University of Health Sciences, Tilzes Str. 18, LT-47181 Kaunas, Lithuania; arnoldas.pautienius@lsmu.lt (A.P.); juozas.grigas@lsmu.lt (J.G.); arunas.stankevicius@lsmu.lt (A.S.); 2Institute of Microbiology and Virology, Lithuanian University of Health Sciences, Tilzes Str. 18, LT-47181 Kaunas, Lithuania; 3Department of Biology, Faculty of Natural Sciences, Vytautas Magnus University, K. Donelaicio Str. 58, LT-44248 Kaunas, Lithuania; marina.sidorenko@vdu.lt (M.S.); jana.radzijevskaja@vdu.lt (J.R.); algimantas.paulauskas@vdu.lt (A.P.)

**Keywords:** tick-borne encephalitis, TBEV RNA prevalence, wild rodents, cell culture

## Abstract

Wild rodents are considered to be one of the most important TBEV-amplifying reservoir hosts; therefore, they may be suitable for foci detection studies. To investigate the effectiveness of viral RNA detection in wild rodents for suspected TBEV foci confirmation, we trapped small rodents (*n* = 139) in various locations in Lithuania where TBEV was previously detected in questing ticks. Murine neuroblastoma Neuro-2a cells were inoculated with each rodent sample to maximize the chances of detecting viral RNA in rodent samples. TBEV RNA was detected in 74.8% (CI 95% 66.7–81.1) of the brain and/or internal organ mix suspensions, and the prevalence rate increased significantly following sample cultivation in Neuro-2a cells. Moreover, a strong correlation (r = 0.88; *p* < 0.05) was found between the average monthly air temperature of rodent trapping and the TBEV RNA prevalence rate in cell culture isolates of rodent suspensions, which were PCR-negative before cultivation in cell culture. This study shows that wild rodents are suitable sentinel animals to confirm TBEV foci. In addition, the study results demonstrate that sample cultivation in cell culture is a highly efficient method for increasing TBEV viral load to detectable quantities.

## 1. Introduction

Tick-borne encephalitis (TBE) is the most common and medically important tick-borne viral zoonosis in Europe and Northern Asia [1]. In Lithuania, the TBE infection rate has been the highest in Europe for nearly 10 years [2]. Although TBE is typically characterized as a focal infection, the whole country of Lithuania is considered to be TBEV-endemic [3].

The tick-borne encephalitis virus (TBEV) maintains its cycle by circulating between vector ticks and reservoir animals [4]. Although the exact mechanism of focal TBEV distribution in nature is still unclear, interactions between ticks and reservoir hosts of the virus, as well as particular environmental and climatic factors, are considered to be the most influential [5,6,7]. A study conducted by Daniel et al. [8] suggests that warm climatic conditions might be related to a higher TBEV load in ticks in the summer and autumn periods and, therefore, to an increased risk of human TBE cases. Moreover, an increased rate of outdoor activities in potentially tick-infested areas in summer and autumn might be related to a higher incidence of human TBE cases [9,10].

Ticks were analyzed for viral RNA presence in the majority of TBEV foci detection studies, although the reported TBEV prevalence rate in ticks was primarily low. Furthermore, a large sample size of ticks must be analyzed to detect TBEV, and tick collection is a time-consuming and labor-intensive task fraught with the risk of tick bites and tick-borne infections [11]. Domestic animals of various species, including dogs, horses, goats, sheep, and wild animals such as rodents, insectivores, roe deer, wild boar, and wild birds, have all been investigated for TBEV seroprevalence and the suitability of TBEV foci monitoring [12,13,14].

To date, only small mammals, such as some species of rodents and insectivores, are suspected to be sufficient reservoir hosts suitable for virus amplification and maintenance [13]. Few studies have shown a correlation between TBEV seroprevalence in rodents and local incidence of human TBE [15,16]. Small rodents might be useful sentinels for TBEV foci detection and monitoring since they tend to be highly infested with ticks, do not migrate long distances, tend to populate in high quantities, and are convenient to trap and collect [15]. Moreover, studies in highly endemic Siberian regions suggest that in endemic areas, the majority of wild rodents might be persistently infected with TBEV [17]. Although the majority of TBEV research in Europe was focused on detecting TBEV-specific antibodies in wild rodents, some authors suggest that the TBEV-Eu subtype can persist in naturally infected rodents without detectable antibodies [18,19].

The present study aimed to investigate the prevalence rate of TBEV RNA in wild rodents in TBE foci in Lithuania. Based on data suggesting persistent TBE infection in rodents, we hypothesized that cultivation of rodent brain and internal organ mix suspensions in murine neuroblastoma Neuro-2a cell culture could increase the viral load to a detectable level and possibly provide more insight into TBEV prevalence and maintenance in wild rodents.

## 2. Materials and Methods

### 2.1. Sample Collection and Sampling Sites

Wild rodents were collected in 19 endemic locations (5 districts) in Lithuania. Specific sites for rodent trapping were selected according to the National Public Health Center under the Ministry of Health, which has provided a complete map of confirmed human TBE cases with probable TBEV-infected tick bite locations or known TBEV foci from 2016 to 2018 (https://nvsc.lrv.lt/lt/uzkreciamuju-ligu-valdymas/uzkreciamosios-ligos/erkiu-pernesamos-ligos/lietuvos-vietoviu-kuriose-uzsikreciama-erkiu-platinamomis-ligomis-zemelapis/ (accessed on 27 February 2024). Moreover, TBEV was detected in questing ticks in the majority of rodent trapping sites in a previous study [20]. Small rodents (*n* = 139) were trapped in different TBEV foci from March 2019 until May 2020, excluding the summer months when trapping was not productive. Rodents were caught in live traps and presented to the laboratory on the same day. The species, sex, and size of the animals were determined, and necropsy was performed according to standard protocols. Brain, spleen, liver, heart, kidneys, and 5 fetal samples were collected and stored at –80 °C until further analysis. The data on average monthly air temperature was obtained from annual reports of the Lithuanian Hydrometeorological Service under the Ministry of Environment (https://www.meteo.lt/category/menesio-hidrometeorologiniu-salygu-apzvalga/page/5/ (accessed on 27 February 2024). The trapping of rodents was performed after confirmation by the Ministry of Environment of the Republic of Lithuania with license No. 5 (28 February 2019) and No. 12 (27 February 2020) according to animal welfare regulations.

### 2.2. Detection of TBEV RNA and Viral Load Quantification

Pieces (10–100 μg) of the brain (*n* = 137) and internal organs such as the spleen, liver, heart, and kidneys of each rodent (*n* = 139) were homogenized separately using a mortar and pestle, then mixed with 1000 μL of Modified Eagle’s Medium (MEM, Gibco, Waltham, MA, USA) and stored at –80 °C for further analysis. Brain samples were analyzed separately from the internal organ mix samples obtained from the same rodent. Total RNA was extracted from 300 μL homogenate with GeneJET RNA Purification Kit (Thermo Scientific, Waltham, MA, USA) according to the manufacturer’s instructions. For the presence of TBEV RNA, extracted samples were analyzed by real-time reverse transcription polymerase chain reaction (RT-PCR) using a reaction mix containing SuperScript™ III One-Step RT-PCR System with Platinum™ Taq DNA Polymerase (Thermo Scientific, Waltham, MA, USA) and primers as described previously [21]. Selected RT-PCR-positive samples were screened by nested PCR using primers and DreamTaq Green PCR Master Mix as described previously [22]. PCR-positive samples were confirmed by partial genome sequencing targeting the 126 bp NCR region of TBEV using the same primer set according to a previous study [21].

The viral quantification assay was modified according to previous studies [21,23]. Briefly, a synthetic fragment corresponding to the amplified TBEV NS5 region was cloned into the pJET1.2 vector using the CloneJET PCR Cloning Kit (Thermo Scientific, Waltham, MA, USA) and transformed into pretreated *Escherichia coli* cells using the Transform-Aid Bacterial Transformation Kit (Thermo Scientific, Waltham, MA, USA) according to the manufacturer’s protocol. Transformed *E. coli* was cultivated overnight at 37 °C. Plasmid DNA was extracted and purified using the GeneJET Plasmid Miniprep Kit (Thermo Scientific, Waltham, MA, USA) and quantified using the Qubit dsDNA BR Assay Kit (Invitrogen, Waltham, MA, USA) according to the manufacturer’s instructions. Standard curves were generated after 10-fold dilutions of stock DNA, which served as templates for qPCR reactions. Sample concentration was calculated using a calibrated standard curve, and viral load was estimated using a quantification assay. Quantification was based on RT-PCR using SYBR Green I Dye (Thermo Scientific, Waltham, MA, USA) and nested NS5 primers as described previously [21]. All samples were tested in triplicates, and mean values were calculated.

### 2.3. Virus Isolation

For cell culture inoculation, 500 μL of homogenate was centrifuged at 12,000× *g* for 10 min and filtered with a 0.22 μm pore size microfilter (Techno Plastic Products AG, Trasadingen, Switzerland). Murine neuroblastoma cells (Neuro-2a ATCC No. CCL-131) were seeded in 96-well plates and incubated in a maintenance medium containing ½ Dulbecco’s Modified Eagle’s Medium (DMEM, Gibco, USA) and ½ Modified Eagle’s Medium (MEM, Gibco, USA) supplemented with 10% heat-inactivated fetal bovine serum (FBS; Gibco, USA), 100 U/mL penicillin, and 100 μg/mL streptomycin at 37 °C in 5% CO_2_. The following day, cells were inoculated for 1 h with 100 μL of microfiltered internal organ mix supernatant and brain supernatant. Cells were then washed with PBS and incubated at 37 °C in 5% CO_2_ in 200 μL of Dulbecco’s Modified Eagle’s Medium (DMEM, Gibco, USA) supplemented with 10% heat-inactivated fetal bovine serum (FBS; Gibco, USA) and 100 U/mL penicillin and 100 μg/mL streptomycin. Cells were inoculated in triplicate for each sample, incubated for 4 days, and monitored for occurrence of cytopathic effect. Following incubation, cells were frozen at –40 °C, thawed two times, and centrifugated at 12,000× *g* for 10 min before the next inoculation. After 3 serial passages, cells were frozen and thawed two times, and the cell suspension was harvested for RNA extraction. Successful virus cultivation was confirmed by RT-PCR.

### 2.4. Statistical Analysis

The exact binomial method was used to calculate confidence intervals. Binary logistic regression analysis, Chi-square test, and Fisher’s exact test were utilized to test the significance of the risk factors. Linear associations were assessed by the Pearson correlation test. All statistical analysis and mapping were performed using the programming language R.

## 3. Results

### 3.1. TBEV RNA Prevalence in Wild Rodents

In total, 137 brain and 139 internal organ mix homogenates from 139 wild rodents were investigated by RT-PCR. TBEV RNA was detected in 104 (74.8%; CI 95% 66.7–81.1) wild rodents. Brain suspensions of 51.1% (CI 95% 42.4–59.7) and internal organ mix suspensions of 53.2% (CI 95% 44.6–61.7) rodents were positive for viral RNA. Furthermore, 28.8% (CI 95% 21.4–37.1) of both brain and internal organ mix suspensions of the same animal were positive for viral RNA. The specificity of the obtained PCR-positive samples was confirmed by partial genome sequencing targeting the 126 bp NCR region (Appendix A Appendix A).

The viral RNA detection rate increased significantly after cultivation in Neuro-2a cell culture (Figure 1), resulting in a total of 134 (96.4%; CI 95% 91.8–98) positive rodents. Brain isolates of 85.4% (CI 95% 78.4–90.9) and internal organ mix isolates of 81.3% (CI 95% 73.8–87.4) rodents were positive for viral RNA. In 69.1% (CI 95% 60.7–76.6) of rodents, TBEV-specific RNA was concurrently found in both types of samples. Moreover, TBEV was detected in both types of rodent samples (*n* = 30), which were PCR-negative before sample cultivation in cell culture.

In addition, viral genome was detected in two out of five fetus suspension samples and in all five samples after isolation in cell culture. In four out of five cases, viral RNA was not detected in relative maternal brain and internal organ suspension samples before isolation in cell culture.

Viral RNA was detected in at least one of the brain or internal organs mix suspension samples of rodents collected in all 19 endemic locations of our study (Figure 2). Based on PCR results from suspension samples, infection rates at the spatial scale ranged from 25 to 100% (Figure 2A). The prevalence rate following sample cultivation in cells increased to 100% in 16 trapping locations, and the adjusted prevalence rate was 71.4–90% in the remaining 3 locations (Figure 2B).

### 3.2. Comparison of TBEV Viral Load in Suspensions and Neuro-2a Cell Isolates

Viral load in the majority of brain and internal organ mix suspension samples was low, and in 70.1% of TBEV-positive suspension samples, amplification was observed at threshold cycle (Ct) values of 35–40. Brain and internal organ mix suspension samples (*n* = 30) with the lowest Ct value and respective rodent sample isolates in cells (*n* = 30) were taken, and viral load was measured (Figure 3). The average viral load in the same rodent’s Neuro-2a cell culture isolates was significantly higher (*p* < 0.05) than in the respective tissue suspension samples (6.4 log_10_ copies/mL and 5.7 log_10_ copies/mL, respectively). The results demonstrate successful TBEV isolation in cell culture and confirm viable virus presence in rodent tissue samples.

### 3.3. Seasonal Trends of TBEV Prevalence in Rodents

Rodent samples were collected when the average air temperature was below 10 °C, and tick activity was not at its peak. The prevalence rate of TBEV RNA in rodent suspension samples from all collection sites was analyzed when grouped according to the month of trapping (Appendix A Appendix A).

A strong correlation (r = 0.88; *p* < 0.05) was found between average monthly air temperature and TBEV RNA prevalence rate in cell culture isolates of rodent suspensions, which were PCR-negative before cultivation (Figure 4). The detected viral RNA prevalence rate in isolates of brain and internal organ mix samples, which were PCR-negative prior to cultivation, was significantly higher in rodents trapped when the average monthly air temperature was higher than 5 °C.

### 3.4. TBEV Prevalence in Different Rodent Species

Overall, rodents of six species were trapped and investigated in the study: 76 *Apodemus flavicollis*, 29 *Clethrionomys glareolus*, 22 *Microtus arvalis*, 9 *Mus musculus*, 2 *Apodemus sylvaticus*, and 1 *Apodemus agrarius*.

TBEV RNA-positive brain and/or internal organ mix suspension samples were detected in 77.6% (59/76) *A. flavicollis*, 72.4% (21/29) *C. glareolus*, 100% (22/22) *M. arvalis*, 75% (6/9) *M. musculus*, 100% (2/2) *A. sylvaticus*, and 100% (1/1) *A. agrarius*. The prevalence rate of viral RNA in brain and internal organ mix suspension samples was found to be significantly higher (*p* < 0.05) in *M. arvalis* rodents compared to other species. On the other hand, the prevalence of TBEV RNA in suspension samples of *A.flavicollis* was significantly lower compared to other rodent species (Fisher’s exact test *p* = 0.46). However, after cultivation in Neuro-2a cells, no significant differences between rodent species and RNA prevalence rate were found. Only 5 of 139 rodents were negative for TBEV RNA, and all of them belonged to *A. flavicollis* species (Appendix A Appendix A).

Five fetal samples were investigated in the study (not included in the overall rodent count). Viral RNA in *M. arvalis* was detected in both maternal samples and fetal samples before and after cultivation in cell culture. In *C. glareolus*, viral RNA was detected only in maternal suspensions before cultivation in neuroblastoma cells. Notably, TBEV RNA was found in one of the *A. flavicollis* fetal suspension samples but not in the respective maternal suspension samples. In the remaining samples of two *A. flavicollis* rodents, viral RNA in both maternal samples and fetal samples was detected only after cultivation in cell culture.

## 4. Discussion

Our study revealed that in confirmed TBEV foci in Lithuania, viral RNA can be found in the majority of wild rodent brain and internal organ mix samples. TBEV RNA was present in 74.8% of rodent brain and internal organ mix suspensions and 96.4% of rodent samples after sample cultivation in neuroblastoma cells. It is the highest reported TBEV RNA prevalence rate in wild rodents, although reports on TBEV-Eu genome presence in wild rodents are scarce. A study performed in high-risk areas in Hungary detected a TBEV RNA prevalence rate of 4.2% (17/405) in rodent liver samples collected over 7 years [24]. In Germany, viral RNA was found in 15% (21/137) of rodent brain and spleen samples in TBEV-risk areas and 8% (24/304) of rodents captured in non-risk areas [15]. A study conducted by Tonteri et al. [18] in two separate TBEV-Sib and TBEV-Eu endemic areas in Finland reported a TBEV RNA prevalence rate of 16.8% (16/95) in TBEV-Eu endemic zone-collected and 6.3% (5/80) in TBEV-Sib endemic zone-collected wild rodents trapped in late winter. Moreover, the reported prevalence was significantly higher in 2009 compared to 2008 (54.2% (13/24) TBEV-Eu and 23.5% (4/17) TBEV-Sib in 2009 vs. 4.2% (3/71) TBEV-Eu and 1.6% (1/63) TBEV-Sib in 2008), although the sampling sites, rodent species, and trapping time were the same in both years of trapping [18]. In endemic areas of Siberia, authors reported a high TBEV-Sib RNA prevalence rate in wild rodents and insectivores, reaching 61–74% [17,25]. Although the data on TBEV RNA prevalence in wild rodents are scarce, the variety of prevalence rates in endemic areas is high, which indicates that many factors might play a role in affecting the prevalence rate.

Several factors might have contributed to the high TBEV prevalence rate in rodents revealed in our study. The human TBE infection rate in Lithuania is the highest in Europe and considerably higher compared to other European countries where TBEV RNA was detected in rodents [2]. Moreover, specific rodent trapping sites were chosen according to the data of detected TBEV-positive ticks, possible tick bite locations, and related human TBE cases. As concluded by Borde et al. [26], the TBEV transmission cycle occurs in small areas (average size of about 0.5–1 ha) of so-called microfoci, which are stable for decades and usually do not expand or shift. Although the TBEV transmission cycles and their stability in microfocus are still not understood, and so far, environmental models do not provide possible explanations [26], capturing rodents in the specific areas of microfocus might increase the chances of TBEV detection. Furthermore, a greater variety of organ tissues were taken for RNA extraction, and brain and internal organ mix samples of each rodent were analyzed separately by RT-PCR. In addition, a landscape of Lithuania rich in agricultural fields, woodlands, and forests, together with humid climatic conditions, is preferable for ticks and might be related to higher tick density. A substantial part of city parks and green zones are also known as TBEV foci, and the retrospective epidemiological study revealed that 60% of TBE patients were inhabitants of cities and 42% were infected in the living area, which shows high TBE risk even in urban areas [27,28]. Moreover, the agricultural landscape is important for the sustainability of the rodent population of some species, and a low trapped rodent number from late spring to early autumn might be related to food abundance in fields. Furthermore, agricultural fields attract deer, which are important tick hosts and have been suggested to be related to the increased risk of human TBE [29,30]. A recent study conducted in Lithuania revealed that the deer population has been exponentially increasing in the past 20 years [31].

Previous studies in Lithuania showed a low TBEV prevalence rate in field-collected ticks. According to various studies, the reported minimal infection rate (MIR) in ticks ranged from 0.1 to 1.8% [20,32,33]. Nevertheless, recent studies confirmed widespread TBEV distribution in the country based on the prevalence rate of antibodies against TBEV in horses (37.5%) and virus presence in horse serum samples (3.9%), as well as in goat (4.3%) and sheep (4.5%) milk samples collected in endemic localities in Lithuania [34,35]. Moreover, in a recent study, TBEV-specific antibodies were detected in 21.6% and TBEV RNA in 18.6% of randomly collected blood serum samples of dogs residing in the second-largest city in Lithuania. Although the vital virus was isolated in cell culture only from a small portion of PCR-positive dog samples, mainly due to the limited volume of obtained blood serum samples, TBEV was isolated from ticks collected from the dogs. Similar to the present study, the TBEV RNA prevalence rate in ticks increased significantly after sample isolation in cell culture (34.2% before vs. 56.4–60.7% after cultivation in different cell cultures) [36].

To the best of our knowledge, no previous studies have attempted to isolate wild rodent-derived TBEV in murine neuroblastoma cells to investigate the virus prevalence rate. A study with a similar goal was performed in an attempt to isolate TBEV in chicken embryo cell culture from *A. flavicollis* blood samples, although it was unsuccessful [37]. After sample cultivation in murine Neuro-2a cells, the detection rate of TBEV RNA and, thus, the overall prevalence rate increased significantly (*p* < 0.01). Successful virus isolation from PCR-negative rodent samples demonstrated that murine neuroblastoma cells are highly susceptible to rodent-derived TBEV infection. Cell lines of neural origin were found to be highly susceptible to TBEV infection, providing 100- to 10,000-fold higher virus titer than cells of non-neural origin [38]. Experimental studies show that virus adaptation to a specific cell line necessitates serial passages for structural rearrangements that favor viral load increasing to detectable levels [39]. As a result, compatible viral host and cell line origin may have played a significant role in high virus isolation success.

The majority of tissue suspension samples were positive for TBEV RNA, albeit characterized by high Ct values, therefore indicating low viral loads. This finding is in line with previous studies that found low viral copy levels in brain and spleen samples of wild rodents [13,15]. Low viral load in the majority of wild rodents might be due to the longevity of TBE infection. Persisting TBEV was found in the brains of various small rodents and insectivores in several investigations [15,17,18,40,41]. A naturally persistent infection in *Clethrionomys rutilus* was demonstrated to last for up to 10 months [42]. However, we have successfully isolated viable TBE virus in murine neuroblastoma cells from rodent samples that were positive at late RT-PCR cycles and from samples (*n* = 30) that were PCR-negative. Michelitsch et al. [41] successfully isolated viable TBEV in human lung carcinoma A549 cells from experimentally infected bank vole brain and internal organ samples that showed low Ct values (<36.0). The success of virus isolation was related to different TBEV strains. In comparison to the reference Neudörfl strain, the more recently isolated TBEV strains were more effectively reisolated in cells, indicating that virus strain plays a role in successful isolation in cell culture [41]. Moreover, although PCR is considered a highly specific and sensitive method for TBEV RNA detection, a study conducted by Donoso Mantke et al. [43] revealed significant limitations in reliably detecting the virus. Only 2 out of 23 laboratories that took part in the study found TBEV in all of the provided human blood serum samples that had different strains and amounts of the virus [43].

A strong correlation was found between average monthly air temperature and TBEV RNA prevalence rate in cell culture isolates of rodent suspension samples that were PCR-negative before cultivation. TBEV RNA was detected more often in rodents captured at a lower average air temperature than in rodents trapped in warmer weather. When the average air temperature of rodent trapping increased, it led to a lower RNA detection rate in suspensions and an increased detection rate in cell culture isolates of the same rodent samples. It is worth noting that the warmest January since 1961 at the time of sample collection, as well as an exceptionally cold May in Lithuania, might have influenced the findings. To the best of our knowledge, no previous study reported a similar trend. A study with known rodent trapping time conducted in Finland found a higher TBEV-Eu RNA prevalence rate in samples collected during colder winter (54.2%) compared to a warmer one (4.2%), even though the collecting area, trapping month, and rodent species were the same. Moreover, the authors reported a high TBEV-Eu RNA prevalence rate in wild rodents trapped in a specific TBEV focus in late winter, when ticks were not active for a few months. In addition, TBEV-specific antibodies were detected in only 12.5% (2/16) of TBEV-Eu RNA-positive rodents and in 100% (5/5) of TBEV-Sib RNA-positive rodents, which indicates that the majority of TBEV-Eu-infected rodents might not have detectable antibodies in persistent TBE infection [18].

The influence of air temperature on TBEV replication was observed in previous studies on hedgehogs (*Erinaceus roumanicus*). According to Nosek and Grulich [44], the duration of viremia in hedgehogs is indirectly proportional to air temperature, lasting 3–6 days in warm summer and 8–14 days in spring and autumn. Additionally, intense and long-lasting viremia can persist throughout the hibernation period in infected mammals [44]. The opposite trend was suggested in ticks: higher air temperature is associated with increased human TBE incidence rate and potentially higher virus load in ticks [8,45]. A study conducted in the Czech Republic reported that most human TBE cases were recorded when the air temperature was 10–20 °C [46]. Moreover, Korenberg and Kovalevskii [47] suggested that the incidence of clinical TBE most closely corresponds to the actual probability of being bitten by a highly infected tick. Furthermore, a thermosensitive RNA switch has been proposed as significant for virus propagation in ticks [48]. However, other factors such as duration of photoperiod, relative humidity [8,26], age and sex [49], reproductive and immune status [50], and tick burden load on rodents [51], which are also at least partially related to the average air temperature, might be influential. Moreover, a rapid fall in ground-level temperatures in August and October is associated with the synchrony of larval-nymphal questing (and therefore co-feeding) in early spring, which was observed in TBEV foci and not elsewhere [6]. However, to date, there is no robust ecological evidence that co-feeding is important for the stability of TBEV cycles in nature [5].

Early spring is a favorable time for TBEV activation from a latent state in rodent organisms due to a weakened immune response during winter [52]. Various factors, such as reproduction status, food availability, climatic conditions, and various stressors, might impact a host’s immune response to virus replication [53]. An increase in rodent male testosterone levels in the spring is associated with humoral immunity inhibition. An experimental study found that injecting testosterone into male red voles can activate the latent TBE infection [54]. Furthermore, TBEV activation was seen in experimentally infected Syrian hamsters given vincristine, prednisolone, and adrenalin [55].

Vertical TBEV-Eu transmission in wild rodents was demonstrated in our study. TBEV RNA was detected in fetuses of *A. flavicollis*, *C. glareolus,* and *M. arvalis* females. To the best of our knowledge, no previous studies have found TBEV RNA in the fetuses of the aforementioned rodent species. However, vertical TBEV-Sib transmission from infected laboratory mice and *C. rutilus* females has been described previously [56,57]. Vertical virus transmission was also demonstrated in an experimental study where TBEV was detected in up to 90% of experimentally infected red vole (*C. rutilus*) progeny. Moreover, transplacental TBEV transfer to embryos was revealed in naturally infected red voles [56]. In addition, sexual TBEV transmission in laboratory mice, which are not adapted hosts, was reported between infected males and uninfected females [58].

In both brain and internal organ mix suspensions, TBEV RNA was significantly more often found in *M. arvalis* compared to *A. flavicollis* and *C. glareolus*. Contrarily, TBEV RNA prevalence was significantly lower in suspension samples of *A. flavicollis* compared to other rodent species. However, results of virus isolation in cell culture demonstrated that the viral genome was absent in only five rodent specimens of *A. flavicollis* species, thus giving a 100% prevalence rate in other collected species. It is important to note that in our study, the majority (77.3%) of *M. arvalis* rodents were collected in January and March of 2020 when the average air temperature was 2.8–3.3 °C and ticks were not active. Consequently, it remains unclear if rodent species or the season of rodent collection was the influential factor for higher prevalence in both brain and internal organ mix suspension samples of *M. arvalis* before cultivation in cell culture.

A study conducted in the Czech Republic revealed that the abundance of *M. arvalis* voles was a significant factor explaining annual morbidity from TBEV and two other zoonoses [59]. TBEV RNA was detected in 11% of *M. arvalis* rodents collected in a non-risk area in Germany [15] and 6.2% in a high-risk area in Hungary [24]. Moreover, persistent TBEV infection for up to 100 dpi was reported in an experimentally infected *M. arvalis* in a study conducted by Achazi et al. [15]. However, *M. arvalis* voles are not a common rodent species in TBEV studies mainly because their main type of habitat is grasslands and not forests and woodlands, as it is for *I. ricinus* ticks and most common rodents in TBEV studies—*A. flavicollis* and *C. glareolus* [60]. Knap et al. [16] have suggested that *A. flavicollis* rodents might be less susceptible to TBEV infection compared to *C. glareolus*, although it has been implied that the rate of infection in a specific species might depend on the year. Moreover, it was suggested that the virus tropism of TBEV in *Apodemus* mice might be different from that in *Clethrionomys* rodents since the virus was predominantly detected in the spleen and less often in the brain, lungs, and blood clots of *Apodemus* rodents. Furthermore, the reported viral loads in the internal organs of *Apodemus* mice were generally lower compared to *Clethrionomys* voles [16]. In experimental studies, *C. glareolus* was demonstrated to produce higher viremia and higher antibody titers in comparison to the rodents of the *Apodemus* species [61,62]. However, regardless of the significant differences in the prevalence of viral RNA in suspension samples of *M. arvalis* and *A. flavicollis* compared to other species, results after sample cultivation in cell culture suggest that all rodent species included in the present study in TBEV foci were persistently infected with TBEV. This finding is in accordance with the study conducted by Achazi et al. [15], who also did not find a significant viral genome prevalence rate difference between rodent species.

There are a few limitations of our study that need to be addressed. The rodent sample size at different collecting sites and months differed, as did the number of rodents sampled for each species. Therefore, the interpretation of the results might not be completely accurate. Trapping of rodents was unevenly successful in different months and in different locations.

## 5. Conclusions

The present study revealed that in TBEV foci in Lithuania, the majority of wild rodents carry the TBE virus. The obtained results demonstrate that sample cultivation in cell culture is a highly efficient method of increasing viral load to detectable quantities. Moreover, the results suggest a higher chance of detecting TBEV RNA in rodents captured in lower average air temperatures than those suitable for ticks, which might help to reduce the number of rodents that have to be captured and analyzed to detect the virus in suspected TBEV foci.

## Figures and Tables

**Figure 1 viruses-16-00444-f001:**
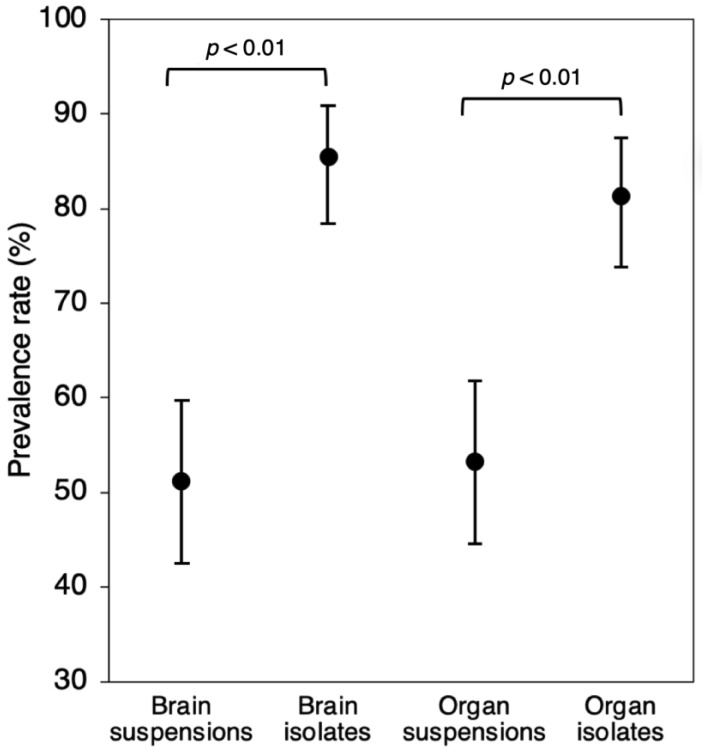
TBEV RNA prevalence rate in wild rodent brain and internal organ mix samples before (suspensions) and after (isolates) cultivation in murine neuroblastoma (Neuro-2a) cell line.

**Figure 2 viruses-16-00444-f002:**
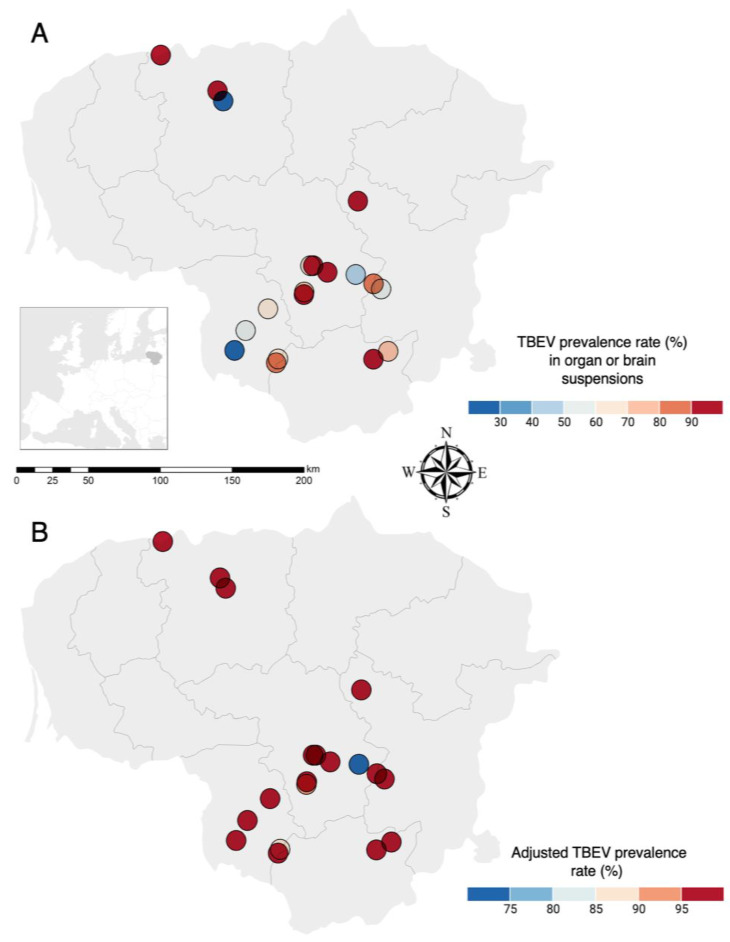
(**A**) Spatial distribution of sample collection sites and average TBEV RNA prevalence rate before isolation and (**B**) after isolation in Neuro-2a cells in different trapping locations in Lithuania. Viral RNA prevalence rate (%) in wild rodents is indicated by different colors shown in legends.

**Figure 3 viruses-16-00444-f003:**
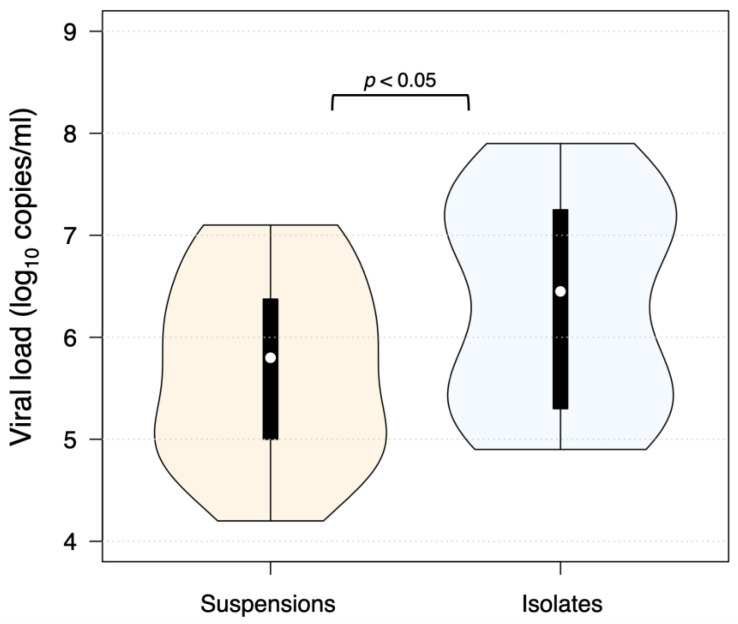
TBEV viral load in selected rodent brain and internal organ mix samples (*n* = 30) before (suspensions) and after (isolates) isolation in murine neuroblastoma Neuro-2a cells.

**Figure 4 viruses-16-00444-f004:**
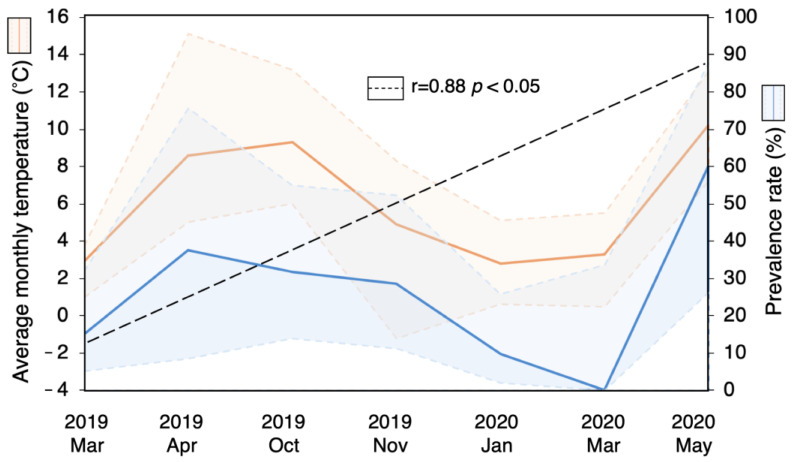
Longitudinal distribution of the prevalence of TBEV RNA in rodent brain and internal organ mix suspension samples that were PCR-negative before isolation in Neuro-2a cell line and the average air temperature of the rodent trapping month.

## Data Availability

Data will be made available on request.

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
