# Peer review of "The Prevalence of Tick-Borne Encephalitis Virus in Wild Rodents Captured in Tick-Borne Encephalitis Foci in Highly Endemic Lithuania"

_viruses, 2024, doi:10.3390/v16030444_

Round 1

Reviewer 1 Report

Comments and Suggestions for Authors

Line 80: How were the species of rodents identified?

Line 145: What is the detection limit of the Real time PCR? 

Figure 4: What does the x axis stand for? The correlation trend line seems to be not suitable since x axis is not either monthly temperature or prevalence.

Author Response

Dear Reviewer,

Thank you for your time and effort in reviewing our manuscript. Please find the responses to your notes below.

Line 80: How were the species of rodents identified?

The species of rodents were identified using standard vertebrae species identification keys, according to Prusaite et al. (1988) and Logminas et al. (1982).

Prusaite J., Mazeikyte R., Pauza D., Pauziene N., Baleisis R., Juskaitis R., Mickus A., Janulaitis Z. (1988) Lietuvos fauna. Žinduoliai. Vilnius “Mokslas”. ISBN 5-420-00055-5.

Logminas V., Prusaite J., Virbickas J. (1982). Vadovas Lietuvos stuburiniams pažinti. Vilnius “Mokslas”.

Line 145: What is the detection limit of the Real time PCR? 

In the present study, suspension samples were still considered PCR-positive when detected at Ct values reaching 40.0. However, other authors report even higher detection limits. In a TBEV study conducted by Andreassen et al. “All real-time RT-PCR positive pools were detected at high Ct values (ranging from 26.4 to 42.4; Median 40.4)”. (https://doi.org/10.1186/1756-3305-5-177)

Figure 4: What does the x axis stand for? The correlation trend line seems to be not suitable since x axis is not either monthly temperature or prevalence.

 The x axis stands for the month of rodent trapping. The x axis was added to Figure 4.

On behalf of all Authors,

Evelina Simkute

Reviewer 2 Report

Comments and Suggestions for Authors

Simkute E., et al: The prevalence of tick-borne encephalitis virus in wild rodents captured in TBE foci in highly endemic Lithuania

This is a well written article - short but potentially impactful. Provided the authors‘ data is correct, it challenges the hypothesis that co-feeding transmission is crucial for TBEV‘s maintenance in nature. As ever, more data is needed, and the article itself should be amended and made more convincing. Specifically, the authors report of an unprecedentedly high, almost absolute prevalence of TBEV infection in murine hosts surveyed with their sensitivity-enhanced method, however, they omit to present any control test demonstrating fidelity of their approach. It comprises too many steps for my tastes - cross contamination might be in play… Absence of controls is particularly critical when data notably departs from what is found in the literature. This is of my biggest concern about this study.  Anti-TBEV antibodies could have been analysed to complete the picture. Also, the authors - in my opinion  overly - accent climate warming as a driving variable in background. Although an impact of growing temperature upon ecosystems in the Baltics is certainly significant, the matter in question concerns, after all, virus’s performance in homoeothermic hosts.

P.1, l.39-41: climate change is an abused hypothesis in attempts to explain TBE rise even in pre-existing disease’s hot-spots (e.g. in Central Europe), but - in terms of ecological niche - once eco-climatic suitability is supreme any change can just lessen it. 

P.2,l.75-6: check the URL address, pls - it returned me the “Page doesn”t exist’ – error message  

P.  2,l.85: ditto

P.3, l.105: according to a previous study

P.3,l,105-15: virus quantification isn’t explained adequately – the authors concentrate on construction  of internal standards rather than the quantification itself.                                                                           

P.3, l.149-p.4,l.150:  explain it better in M&M, pls.

P.6,l.200 and throughout: round to 1-2 decimals, pls                                     

P.6,l.200-5: this passage isn’t clear enough to me, and Fig.4 adds to confusion…  What kind of samples the category „CONSIDERED negative“ ever comprises? It sounds quite subjective, and isn’t defined in M&M.

P.7,l,217-9: „..The prevalence rate of viral RNA in brain and internal organ mix suspension samples was found to be significantly higher (p < 0.05) in M. arvalis rodents, compared to other species…“, if it is so, it is a paradox as the Common vole avoids forests, and occupies open grassland and crop fields which makes him the least tick bite-exposed species of the six…

P.7,l,219-23: note that – if a null that the virus predominance in A. flavicollis is lesser than that in the rest of  species is tested – significance level in Fisher exact test approaches 5% (p=0.46, upper tailed test) – in my, opinion it is worth of discussion…

P.8,l.310: mixing “Ct” and “Cq” in a text is confusing – choose only one, pls.

P.9,l.319-27: the observed pattern of concurrence needn’t necessarily imply causality: spontaneous fluctuations in TBEV occurrence are notorious, and the period of observation is too short to rule their effect out.

P.9, l.342-3: “..The opposite trend was observed in ticks: higher air temperature is associated with in-  creased human TBE incidence rate and higher virus load in ticks [8]…” - be precise when citing a source, pls. – in [8], no data evidencing „higher virus load in ticks“ is shown.

P.9, l.361: delete „white“, pls.

P.9, l.366: rodent specimens

Fig. 4: horizontal scale is missing! What does “Longitudinal distribution of rodent brain“mean?  Is it measured from frontal to occipital lobe, or from East to West? Both is senseless…

Table S2: note it is meaningless to georeference localities with sub-metre precision – round the Lon/Lat’s to a reasonable number of decimals, pls  (6 decimal digits ~ 11cm, 5 digits ~ 1m, 4 digits ~ 10m, …)

Author Response

Dear Reviewer,

Sincerely,

Evelina Simkute

Round 2

Reviewer 2 Report

Comments and Suggestions for Authors

Simkute E., et al: The prevalence of tick-borne encephalitis virus in wild rodents captured in TBE foci in highly endemic Lithuania V2.

I acknowledge that the manuscript has been amended according to (some of) my suggestions. Unfortunately, some other issues have emerged and should be corrected.

P.3, l.103: according to previous studies [21,23]

P.6, l.106 and throughout:   although assuring of improvement in their response, the authors continue in reporting results of statistical testing with undue number of decimals. Perhaps, they worry about a loss of information… Actually, such an obsession about a thousandth on a probability scale - when observations are few - isn’t adverse but ‘clumsy’.

Throughout the list of references and the text: in an effort to reinforce their argumentation, the authors have added a number of references (to me in surplus to what would be an optimum), which threw the index system into disorder - as a result, for example, references indexed from 64 on, though on the list, are not linked with the text. A careful revision is needed.

Author Response

 Dear Reviewer,

Thank you for your time and effort in reviewing our manuscript and providing additional notes. Please find the responses to your notes below.

P.3, l.103: according to previous studies [21,23]

Corrected.

P.6, l.106 and throughout although assuring of improvement in their response, the authors continue in reporting results of statistical testing with undue number of decimals. Perhaps, they worry about a loss of information… Actually, such an obsession about a thousandth on a probability scale - when observations are few - isn’t adverse but ‘clumsy’. 

Numbers of decimals were rounded in Fig. 1, Fig. 4, and throughout the text.

Throughout the list of references and the text: in an effort to reinforce their argumentation, the authors have added a number of references (to me in surplus to what would be an optimum), which threw the index system into disorder - as a result, for example, references indexed from 64 on, though on the list, are not linked with the text. A careful revision is needed.

Thank you for noting. References and their indexes in the text and list were carefully revised and some of the references were removed.

On behalf of all Authors

Evelina Simkute